# Arbuscular Mycorrhizal Fungal Communities of Native Plant Species under High Petroleum Hydrocarbon Contamination Highlights *Rhizophagus* as a Key Tolerant Genus

**DOI:** 10.3390/microorganisms8060872

**Published:** 2020-06-09

**Authors:** Soon-Jae Lee, Mengxuan Kong, Marc St-Arnaud, Mohamed Hijri

**Affiliations:** 1Department of Ecology and Evolution, University of Lausanne, 1015 Lausanne, Switzerland; soon-jae.lee@unil.ch; 2Institut de Recherche en Biologie Végétale, Université de Montréal and Jardin Botanique de Montréal, 4101 Sherbrooke est, Montréal, QC H1X 2B2, Canada; cookykmx@gmail.com (M.K.); Marc.St-Arnaud@umontreal.ca (M.S.-A.); 3AgroBioSciences, Mohammed VI Polytechnic University, Lot 660—Hay Moulay Rachid, 43150 Ben Guerir, Morocco

**Keywords:** arbuscular mycorrhizal fungi, community structure, Petroleum-hydrocarbon contamination, PCR, cloning and sequencing, Ribosomal RNA, extreme environment, tolerance, *Rhizophagus*

## Abstract

Arbuscular mycorrhizal fungi (AMF) have been shown to play an important role in increasing plant fitness in harsh conditions. Therefore, AMF are currently considered to be effective partners in phytoremediation. However, AMF communities in high levels of petroleum pollution are still poorly studied. We investigated the community structures of AMF in roots and rhizospheric soils of two plant species, *Eleocharis elliptica* and *Populus tremuloides*, growing spontaneously in high petroleum-contaminated sedimentation basins of a former petrochemical plant (91,000 μg/Kg of C10–C50 was recorded in a basin which is 26-fold higher than the threshold of polluted soil in Quebec, Canada). We used a PCR cloning, and sequencing approach, targeting the 18S rRNA gene to identify AMF taxa. The high concentration of petroleum-contamination largely influenced the AMF diversity, which resulted in less than five AMF operational taxonomical units (OTUs) per individual plant at all sites. The OTUs detected belong mainly to the Glomerales, with some from the Diversisporales and Paraglomerales, which were previously reported in high concentrations of metal contamination. Interestingly, we found a strong phylogenetic signal in OTU associations with host plant species identity, biotopes (roots or soils), and contamination concentrations (lowest, intermediate and highest). The genus *Rhizophagus* was the most dominant taxon representing 74.4% of all sequences analyzed in this study and showed clear association with the highest contamination level. The clear association of *Rhizophagus* with high contamination levels suggests the importance of the genus for the use of AMF in bioremediation, as well as for the survey of key AMF genes related to petroleum hydrocarbon resistance. By favoring plant fitness and mediating its soil microbial interactions, *Rhizophagus* spp. could enhance petroleum hydrocarbon pollutant degradation by both plants and their microbiota in contaminated sites.

## 1. Introduction

Arbuscular mycorrhizal fungi (AMF) are obligatory fungal symbionts forming a symbiosis with up to 80% of land plant species on earth [1,2]. As mutualists, AMF generally improve plant growth by increasing their uptake of mineral nutrients, in particular, phosphorus [3,4,5]. As a trade-off, AMF receive carbon from the plant [6]. In addition to their role in plant nutrition, multiple studies have shown that AMF enhance plant survival from biotic and abiotic stress such as nutrient limitations, fungal and bacterial plant pathogens, plant-parasitic nematodes, salinity, drought, trace elements and petroleum hydrocarbon pollutants [7,8,9,10,11,12,13,14]. The crucial roles of AMF in the survival of host plants in harsh environments have led researchers to consider them as useful partners in phytoremediation. To date, a number of studies have shown that AMF could improve phytoremediation processes to clean up soil polluted with trace metals [8,15,16,17]. Many studies have also indicated that the application of AMF can impact the community structure of soil microbes that enhance degradation, sequestration or stabilization of pollutants [18,19].

However, the mechanisms through which AMF can favor phytoremediation are yet to be understood. Moreover, multiple variables can affect the AMF community structure, especially in field conditions. AMF were shown to exhibit host-plant dependency and a community structure that varied in different environmental conditions [20,21,22,23]. Previous studies also indicated that the diversity of AMF could be modified by contaminants such as trace elements and petroleum hydrocarbons [19,24,25,26]. These various factors affect AMF community structure and make the application of AMF in the field challenging and unpredictable. In addition, studies on remediation of petroleum hydrocarbons by the use of AMF in large-scale trials remain to be conducted. By better understanding the dominant AMF taxa in petroleum hydrocarbon contaminated sites, we could harness these taxa, which potentially exhibit high tolerance to contamination stress, and use them as bioinoculants in phytomanagement. Therefore, it is essential to increase our knowledge of the effect of biotic and abiotic environmental factors on AMF communities in highly petroleum hydrocarbon contaminated sites to design successful phytoremediation strategies.

At the same time, multiple genome sequencing projects on AMF suggest different phylogenetic clades of AMF can have dramatically different gene repertoires for their adaptation and functioning in ecosystems [27,28]. Thus, if there is strong phylogenetic signal observed among AMF under high contamination, it is likely that the conserved genes of certain clades are related to the distribution. To date, it has not been clearly understood whether certain AMF clades exhibit high tolerance against high petroleum hydrocarbon contamination. Many papers on AMF have reported the occurrence of members belonging to the genus *Rhizophagus* [24,26], which was considered to be an important genus because of its dominance in both inorganic and organic polluted soils.

The objectives of this study were (1) to investigate the diversity and community structures of AMF associated with two native plant species growing spontaneously in high petroleum hydrocarbon polluted environments; and (2) to evaluate the effect of contaminant concentrations on AMF community structure. Specifically, we hypothesized that (1) plants recruit different AMF communities under different concentrations of petroleum hydrocarbon, and those communities exhibit some dominant taxa, particularly in highly contaminated sites; (2) contamination concentration shapes AMF community structure much more strongly than host identity and biotopes. To address these objectives and test the hypotheses, we used PCR, cloning and Sanger sequencing based on the 18S rRNA gene to amplify an approximately 750 bp fragment from AMF. We report the correlations between the AMF taxa and three distinct environmental factors: host plant species identity, biotopes (roots or rhizospheric soils), and contaminant concentrations (lowest, intermediate and highest).

## 2. Materials and Methods

### 2.1. Site of Study and Sampling

Sampling was conducted in artificial sedimentation basins of a former petrochemical plant located in Varennes, Montreal region, Quebec, Canada (45°41′56″ N 73°25′43″ W), where petroleum hydrocarbon wastes were dumped for several decades. Details on the site of study were previously published [18,29,30]. An exhaustive vegetation inventory [31] was conducted in one of the decantation basins sampled in the present experiment, where the authors described the site as being patchy revegetation dominated by *Eleocharis obtusa* and *Panicum capillare*. However, the plant diversity differed between the basins and this led us to focus on plant species co-occurring in the three basins. Therefore, in this study, we choose to sample the plant species *Eleocharis elliptica and Populus tremuloides*, which were present in all three basins, allowing us to compare their related AMF community structures. We collected three randomly chosen individual plants for each plant species in each basin. For each plant, roots and the rhizospheric soil were sampled. *Populus tremuloides* is a tree species and its individuals were approximately one or two years old, while *Eleocharis elliptica* is herbaceous. Samples were separately sealed in plastic bags, stored in a cooler filled with icepacks, and immediately transported to the lab. Taxonomic identification of plants was conducted with partial specimens, which did not affect the further harvesting of root samples. Specimens of each species were deposited in the Marie-Victorin herbarium (Biodiversity Center, Institut de recherche en biologie végétale, Université de Montréal, Montreal, QC, Canada). Roots were separated from the rhizospheric soil and carefully washed under tap water. Sterile water was used for the final washing step and samples were immediately stored at −80 °C until the DNA extraction. Rhizospheric soils were also collected and stored at −80 °C.

### 2.2. DNA Extraction

Root DNA was extracted using the commercial DNeasy Plant Mini kit (Qiagen, Toronto, ON, Canada) from 100 mg root subsamples crushed with liquid nitrogen using a mortar and pestle. Soil DNA was extracted from 500 mg soil subsamples using the PowerSoil DNA Isolation kit (MoBio Laboratories-Qiagen, Toronto, ON, Canada). Both extractions series were performed following the manufacturer’s instructions.

### 2.3. PCR, Cloning and Sequencing

PCR amplifications were individually performed on the DNA extracted from the root and soil samples using primer pair AML1 (5′-ATCAACTTTCGATGGTAGGATAGA-3′) and AML2 (5′-GAACCCAAACACTTTGGTTTCC-3′) to amplify a 750 bp fragment of the 18S rRNA gene [32]. PCRs were performed using the following cycling program: initial denaturation at 94 °C for 3 min, followed by 30 cycles at 94 °C for 45 s, 55 °C for 45 s, 72 °C for 45 s, and a final extension period at 72 °C for 10 min. One µL of diluted (1/10) DNA was used as a template for PCR reactions in a 50 µL volume containing: 1× PCR buffer, 1U of *Taq* DNA polymerase (Qiagen, Toronto, ON, Canada), 0.25 mM dNTP mixture, and 0.4 µM of each primer. PCR products were run on a 1% agarose electrophoresis gel, stained with GelRed, and visualized using a GelDoc imaging system (Bio-Rad, Mississauga, ON, Canada). The PCR products were cloned using the CloneJET PCR Cloning kit (Thermo Fisher Scientific, Mississauga, ON, Canada) following the manufacturer’s instructions. Ligated plasmids were transformed into competent *Escherichia coli* TOP10 cells (Thermo Fisher Scientific, Mississauga, ON, Canada) using a heat-shock approach. The transformed bacteria were plated onto LB (Luria–Bertani) medium containing 100 µg/mL ampicillin. PCR using AML1 and AML2 primers was performed directly on bacterial colonies to screen positive clones. A total of 976 clones that showed fragments with the expected size were sent for sequencing using a commercial service provided by the Genome Quebec Innovation Center (Montréal, QC, Canada).

### 2.4. Soil Contaminant Analyses

Composite soil samples were taken from nine soil subsamples (from the root zone of each of the three replicates per each of the two analyzed plant species and an additional non-mycorrhizal plant species (*Persicaria maculosa*) not analyzed further in the present article) in each basin where plants were sampled on 19 July 2012. The composite soil samples from each basin were characterized by measuring the sum of all aromatic and aliphatic petroleum hydrocarbons with chain lengths of C10–C50 and polycyclic aromatic hydrocarbons (PAH) using a commercial service provided by Maxxam Analytics, Montréal, Quebec, Canada. The concentrations of these petroleum hydrocarbons are shown in Appendix A. The total petroleum hydrocarbon (C10–C50) concentrations of the three basins were 3000 µg /kg, 41,000 µg /kg and 91,000 µg /kg, and total PAH values were 8.4 µg/kg, 5220 µg/kg and 7000 µg/kg, which were categorized respectively as Lowest concentration (LC), intermediate concentration (MC) and highest concentration (HC) among the three polluted sites. The total petroleum hydrocarbon (C10–C50) concentration of 91,000 μg/Kg is 26-fold higher that the threshold of polluted soil for industrial use in Quebec Province, Canada.

### 2.5. Bioinformatics and Statistical Analyses

Sequences were examined and trimmed using Mothur (v.1.31.2) [33]. Among the 1074 rRNA gene sequences recovered from Sanger sequencing, 98 sequences were excluded in our analyses because they had poor sequencing quality or they were represented only by one sequence (forward or reverse) which was shorter than the expected length (750 bp). A similarity threshold of 97% (uncorrected pairwise distance) was used for SSU (small subunit) sequences belonging to the same operational taxonomic unit (OTU). BLAST [34] was conducted in the curated MaarjAM database [35] with the consensus sequences of each OTU as queries. All sets of the close sequences exhibiting at least 97% of sequence similarity with each query were retrieved and further combined with the well-defined reference sequences from Kruger et al. [36]. Multiple sequence alignment was conducted using MUSCLE v.3.6 [37]. The DNA substitution model was determined using the Bayesian information criterion calculations implemented in jModelTest v.2.1.7 [38]. Bayesian phylogenetic analyses were performed with 20,000 generations of trees and the first 3000 trees were removed. Rarefaction analysis, Bray–Curtis dissimilarity, alpha diversity calculations (Shannon, Simpson and inverse Simpson indices) and permutation analysis of variance (PERMANOVA) were conducted in R (v.3.3.2) using the vegan package. Welch’s ANOVA followed by the Games–Howell post-hoc test was applied with the default function in R for the comparison of richness of samples and abundance of individual OTUs. The package phyloseq was used to undertake principal coordinate analysis (PCoA) and distance-based redundancy analysis (db-RDA). All sequences related to this project were deposited in GenBank database under accession numbers: MF788214–MF789352.

## 3. Results

### 3.1. AMF Molecular Identity and Diversity

In total, 1074 clones were analyzed and provided 976 quality-controlled sequences that were initially assigned to 36 operational taxonomic units (OTUs) based on the 97% level of sequence similarity (Appendix A). Of 36 OTUs, 27 OTUs were singletons or doubletons, and were excluded from further analyses. On the contrary, nine OTUs were well represented by more than two sequences and targeted for in-depth analyses. AMF sequences were detected in both root and soil samples of *Eleocharis elliptica* and *Populus tremuloides*, of which families have been reported as mycorrhizal hosts [18,26].

We initially investigated the number of OTUs related to each host-plant species with regard to the concentration of contamination (Table 1). It has been reported that contamination could greatly reduce the richness of AMF in the community [24,39]. Interestingly, the results of our study in the highly polluted basin suggested this is not necessarily true in all cases (Figure 1). In both root and soil samples of *E. elliptica*, the total number of OTUs showed a decreasing pattern with increasing contamination concentration. On the contrary, observations differed in *P. tremuloides*, as there was no such clear pattern observed. However, there was no significant differences between contamination levels due to the high variance between samples (*p* > 0.05). The rarefaction curves of each sample reached saturation for most cases, although the number of OTUs detected in each sample was equal to or less than four (Appendix A). The rarefaction analysis indicated that the retrieved sequence dataset of each sample was sufficient to represent the AMF communities.

BLAST-based identification of OTUs was followed by a Bayesian phylogenetic analysis to confirm and summarize the taxonomic identity of the nine OTUs (Appendix A and Appendix A). The result of the Bayesian phylogenetic analysis agreed well with the BLAST-based identification. AMF OTUs were assigned into five families (Glomeraceae, Claroideoglomeraceae, Acaulosporaceae, Diversisporaceae and Paraglomeraceae) and six genera (*Rhizophagus, Claroideoglomus, Acaulospora, Diversispora, Paraglomus* and *Funneliformis*). The family Glomeraceae was dominant compared to the other families and it was represented by two genera (*Rhizophagus* and *Funneliformis*) and three virtual taxa (VTX00067, VTX00113, and VTX00114), while Acaulosporaceae and Paraglomeraceae were only represented by one OTU each. Overall, OTU1 (*Rhizophagus irregularis*, VTX00114) was the most dominant species, comprising 71% of all sequences analyzed. The second dominant taxon was OTU2 (*Claroideoglomus* sp., VTX00193), which represented 14.2% of the sequences, while the remaining seven OTUs were represented by less than 10% each relative to the total sequences.

Shannon and Simpson indexes of the AMF communities are shown in Table 1. As expected with the number of OTUs detected, the observed values of Shannon index fell between 0 and 1.1 in all cases, which was relatively low compared with the diversity found in other studies [25,26,40]. A similar low diversity was calculated with the inverse Simpson index. There was no significant difference found for all three alpha diversity indices between different contamination levels, host plant identity and biotopes (root or soil).

The relative abundances of OTUs are summarized in Figure 2 considering three factors: contamination concentrations, host-plant species identity and biotopes. With its largest proportion of total sequences, OTU1 (*Rhizophagus irregularis*, VTX00114) was the most abundant AMF colonizing the roots in general. OTU1 was especially dominant in root samples of *E. elliptica* by taking over 90% of the abundances regardless of the contamination concentrations. It was also frequently found in the rhizospheric soil of *E. elliptica* with percentages of 91.7% (LC), 78.7% (MC), and 98.1% (HC). OTU1 was also dominant in the root samples of *P. tremuloides* under LC and HC with percentages higher than 85%, but only represented 16.7% of the sequences in the case of MC, where OTU3 (*Acaulospora* sp., VTX00028) accounted for 79.2% of the sequences. The general dominance of OTU1 was not observed in the rhizospheric soil samples of *P. tremuloides*, where OTU2 (*Claroideoglomus* sp., VTX00193) was the dominant taxon with a percentage of 45.8% in LC, 70.9% in MC, and 59.1% in HC basins.

### 3.2. AMF Community Structure

To summarize the general community differences, a principal coordinate analysis (PCoA) was performed with Bray–Curtis dissimilarity indices (Figure 3). PERMANOVA was applied for understanding the main effects affecting the AMF community. Strongly significant effects of plant identity and biotope on the AMF community structure were detected (*p* < 0.001, *R*^2^ = 0.2196 (plant identity) and *R*^2^ = 0.1128 (biotope)), while no significant effect of contamination was found (*p* > 0.05, *R*^2^ = 0.1201). Interestingly, the results of PCoA and PERMANOVA discriminated the effect of contamination on AMF communities associated with two plant species. AMF communities associated with *E. elliptica* showed a remarkably low level of variation between samples from different contamination concentrations regardless of biotope, and there was no difference shown by PERMANOVA between the three levels of contamination (*p* > 0.05, *R*^2^ = 0.2045 (root) and *R*^2^ = 0.4640 (soil)). On the other hand, in *P. tremuloides*, AMF communities showed clear differences among three levels of contamination, with the separation following the first principal axis (60.7%). The effect of contamination concentration was significant in root samples (*p* < 0.05, *R*^2^ = 0.7061), while there was no significant effect in soil samples (*p* > 0.05, *R*^2^ = 0.3574).

Finally, a distance-based redundancy analysis (db-RDA) was performed with 1000 Monte Carlo permutations to analyze the relationship of individual AMF OTUs with the three environmental factors: host-plant species, contamination concentrations and biotopes (Figure 4). The permutation test showed constrained ordination was successful (*p* < 0.001). We first found several AMF OTUs showing an association with host plants. OTU1 (*R. irregularis*, VTX00114), OTU4 (*Rhizophagus* sp., VTX00113) and OTU7 (*Paraglomus* sp., VTX00350) were related with *E. elliptica*, while all other six OTUs were more related to *P. tremuloides*. Second, associations between biotopes and specific OTUs were detected. OTU1, OTU4 and OTU7 were mostly found in roots samples, and OTU2 (*Claroideoglomus* sp., VTX00193), and OTU8 (*Funneliformis mosseae*, VTX00067) were more frequently present in soil samples. Third, we could observe the association of certain OTUs with contamination levels. OTU3 (*Acaulospora* sp., VTX00028) and OTU9 (*Diversispora celeta*, VTX00060) were mostly found in the medium contamination concentration, and OTU1 and OTU4 in the high contamination concentration. There was no clear association of OTUs related to the low contamination concentration. Interestingly, we found strong association of OTUs from the same genus along the first axis of db-RDA (36.9%). All three genera that had more than one OTUs in our study, *Rhizophagus*, *Diversispora* and *Claroideoglomus*, showed the close association of OTUs within each genus.

## 4. Discussion

The diversity of plant species found at a site can be strongly influenced by concentrations of inorganic and organic contaminants [41,42,43]. Due to the high concentrations of petroleum hydrocarbon contamination (Appendix A) that greatly affected the diversity and distribution of spontaneous plant species [31], few plant species co-occurred in all three decantation basins targeted in our study, among which were the two selected species *P. tremuloides and E. elliptica*. The soil contamination could also significantly reduce the number of species or modify the respective abundance of AMF species in a community [24,39]. Accordingly, of the 36 AMF OTUs detected overall in this study, only nine were found more than twice, with between one and to four OTUs only in each combination of plant species and contaminant concentration. Interestingly, despite the high level of petroleum hydrocarbon contamination (up to 91,000 µg TPH/kg of soil, which represents 9.1% (w/w) of TPH in soil), we detected AMF either in roots or in the plant rhizospheres that were obviously interacting with plants in this highly contaminated environment. Most of the sequences were formed by OTUs belonging to Glomerales (85.24%), and the remainder (less than 15%) consisted of OTUs from Diversisporales and Paraglomerales. These three orders were also previously reported from the roots of plants growing in high levels of metal contamination [44]. The confirmed occurrence of AMF under high contamination supports the idea that these fungi are effective partners in the detoxification processes and in the alleviation of abiotic stress in plants [8,15,16,17,24,25,40]. The ability of host plant species to strongly influence their AMF communities was previously documented [21,22,23]. However, the effect of petroleum hydrocarbon contamination on the selection of AMF taxa by plants is still poorly known. Here, the changes in AMF community structure were highly correlated with the host plant identity and the biotope (roots and rhizosphere soil). We found the influences of biotopes and hydrocarbon contamination on AMF community structure, with shift patterns differing among host plant species. We found that AMF communities associated with *E. elliptica* were not significantly affected by the concentration of petroleum hydrocarbons in both roots and rhizospheric soils (Figure 3). This result did not support our hypotheses. Contrarily, the AMF communities associated with *P. tremuloides* were clearly affected by contamination concentration, which supports our first hypothesis. Interestingly, AMF community of roots were affected, while soil AMF community were not affected by petroleum hydrocarbon concentration. The finding was unexpected because the soil AMF community should be linked with the root AMF community in general, as AMF are obligatory plant root symbionts. There are three possible explanations for this observation. First, the AMF community of *P. tremuloides* roots were in transition phase when the sampling took place (cessation of oil refining activities by the petrochemical plant), assuming that soil AMF community might remain unchanged overtime during the season. Indeed, the AMF community associated with a host plant can change following the season or the growing stage of plants [26]. A second possible hypothesis is the close association between soil bacterial and fungal community and the AMF community that can increase the stability of the AMF community in soil. As suggested by the concept of plant holobiont, mycorrhizal networks of AMF serve as a backbone for the below-ground components of the holobiont [18,19,45,46]. There are various soil bacteria that have AMF hyphae as their ecological niche. Some of these bacteria can even form biofilm-like structures on the surface of hyphae [47,48]. These is still a lack of knowledge about how deep the intimate association between soil microbial communities and AMF could happen. However, considering the intimacy between AMF and interacting bacterial species, the resilience of AMF and soil microbial communities against environmental changes could come from both. Currently, one of the largest obstacles to applying AMF at contaminated sites is that the AMF community can be affected by the contamination [24,25,26], thus the expected effect from the induced shift of the AMF community by AMF inoculum can be mitigated by the contamination. This second scenario, if true, could be crucial to further developing strategies to remove contamination from soil using the AMF and AMF-interacting microbial community together instead of solely via AMF inoculation. This could be a means of stabilizing the functioning of inoculated AMF or bacterial species by stabilizing AMF and soil microbial community. Further efforts to screen and isolate bacterial species intimately interacting with AMF under high contamination, and to understand the interaction between those bacteria and AMF, should be made. The third putative scenario is the dispersion of AMF communities either by wind or by runoff, or even by birds, from the basin digs where the plants were growing [49,50].

At the same time, previous studies reported that host plants could favor certain AMF species to become dominant in their rhizosphere under particular environmental conditions [20,51]. Furthermore, evidence has been reported in support of a narrowed specificity of effective symbiotic partners due to the dialog response of both AMF and plants, which could exert selection pressure. Examples of selection have been known for a number of decades, such as the intimate AMF plant specificity reported between three legume species, *Medicago sativa*, *Hedysarum coronarium* and *Onobrychis viciaefolia*, and four *Glomus* species when grown in two soils with different phosphorus (P) availability [52]. 

In our study, OTU1 (*R. irregularis*, VTX00114) was the most dominant taxon and represented 71% of all AMF sequences (Figure 2). *R. irregularis* is one of the most common AMF species and is frequently found in diverse ecosystems. It was also reported as the dominant AMF species in various contaminated sites, such as soils contaminated with trace metals and petroleum hydrocarbons [24,25,26]. In our study, *Claroideoglomus* sp. (VTX00193) was the dominant OTU in the rhizosphere soil samples of *P. tremuloides*, except for the samples from the basin with the lowest contamination level, while *R. irregularis* was the dominant OTU in all samples from *E. elliptica* rhizosphere. At the same time, nearly all root samples, regardless of host plant species, did not show any difference in dominant AMF OTUs (except in the case of *P. tremuloides* from MC soil showing *Aculospora* sp. VTX00028 as the dominant species). Instead, they all shared *R. irregularis* as a common dominant AMF OTU. It has been known that AMF OTUs were highly associated with the three environmental factors (contaminant concentration, host plant identity, biotope) (Figure 3 and Figure 4), which is in line with Velazquez, et al. [53], who reported that AMF occurrence under certain environmental conditions varied significantly between species. Not only the overall dominance of *R. irregularis* regardless of the contamination level, but the result of the db-RDA showing high level of association of *R. irregularis* with high contamination concentrations was also in line with previous reports suggesting that *R. irregularis* have a high tolerance in extreme environments. This species was also frequently found in sites contaminated with trace metals and known to mitigate the contamination effect on plants, which suggests the species as a promising partner for bioremediation of petroleum hydrocarbon contamination [24,26]. 

Recently, it was revealed from genome sequencing studies that the functional gene repertoire of different AMF species can vary significantly, implying a functional difference among different clades of AMF [27,28]. If a phylogenetic signal of AMF for their association with contamination can be found, it will suggest that certain common features might be associated with unique genes encoded in the genome of that clade that might be linked to their tolerance or stability against contamination stress. Surprisingly, we found that AMF species of the same phylogenetic clade have similar patterns of distribution against contamination concentrations and host plant identity. We found several OTUs belonging to the genus *Diversispora* (*Diversispora eburnea* and *Diversispora celeta*), *Claroideoglomus* (*Claroideoglomus* sp. (VTX00193 and VTX00276) and *Rhizophagus* (*R. irregularis* VTX00114 and *Rhizophagus* sp. VTX00113)). The db-RDA revealed that *Diversispora* and *Claroideoglomus* showed a tendency to be associated to the host plant *P. tremuloides*, but did not show unified patterns of distribution with contamination concentrations and biotopes. Thus, the conserved gene repertoire between these two genera is not likely to be related to AMF tolerance against petroleum hydrocarbon contamination. On the contrary, both OTUs (VTX00113 and VTX00114) that belong to the genus *Rhizophagus* showed clear correspondence with high contamination. Moreover, species from this genus have been continuously reported to be dominant under various polluted environments including extreme heavy metal contamination [24,26]. The results therefore suggest narrowing the target AMF taxon to the genus level for conducting a functional gene survey to understand the mechanisms of the tolerance against soil contamination.

In summary, the high concentrations of petroleum hydrocarbon contamination considerably decreased the AMF diversity. This high contamination also greatly influenced the number of OTUs found in this study. However, AMF communities were not structured only by the level of petroleum hydrocarbon contamination. We showed that plant identity and biotopes also profoundly affected OTUs abundance and influenced their community structure. Overall, as reported in other studies of site contaminated with trace metals and petroleum hydrocarbons [24,26], *R. irregularis* was also found to be the dominant OTU in the three basins. Moreover, we found a strong association of OTUs of the genus *Rhizophagus* with high levels of contamination. The observed association in congruence with the accumulating reports of tolerance of *Rhizophagus* against abiotic contamination suggests the importance of *Rhizophagus* for future applications of AMF in bioremediation, as well as a survey of key genes for understanding AMF tolerance against trace elements and petroleum hydrocarbon stresses. The outcome of this investigation allowed us to trap and isolate two *Rhizophagus irregularis* strains which were deposited at the Canadian National Mycological Herbarium (DAOM), Ottawa, Canada under the accession numbers 242422 and 242423. The strain DAOM-242422 was used as an inoculant in a phytoremediation trial in the site of a former industrial landfill planted with willows [54]. Further investigations are required for better understanding the role of *Rhizophagus* spp. in anthropized environments.

## Figures and Tables

**Figure 1 microorganisms-08-00872-f001:**
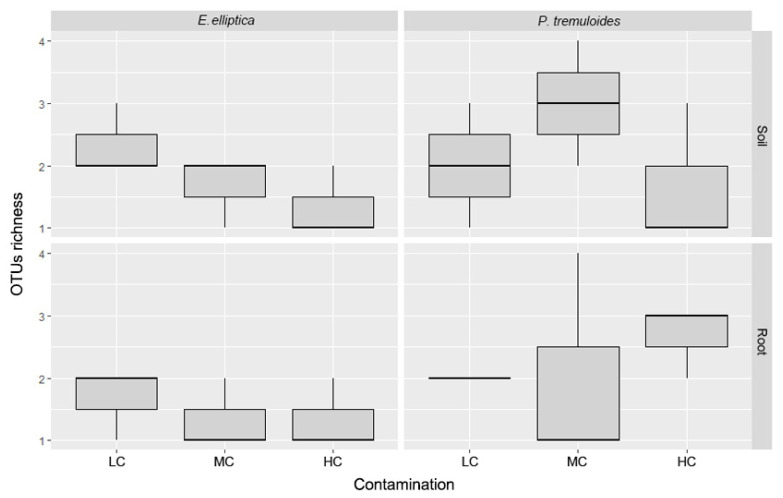
Box plot of observed Glomeromycota OTU richness associated with *P. tremuloides* and *E. elliptica* from decantation basins with lowest (LC), intermediate (MC) and highest (HC) petroleum hydrocarbon contamination. Welch’s ANOVA followed by the Games–Howell post hoc test showed no significant difference among three basins.

**Figure 2 microorganisms-08-00872-f002:**
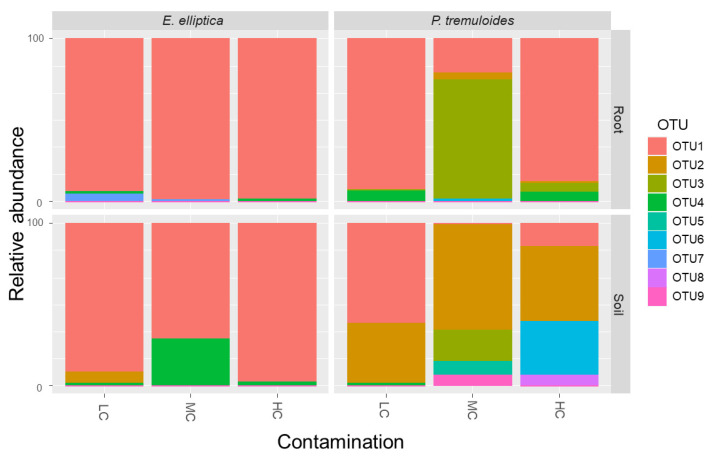
Relative abundance of nine Glomeromycota OTUs in the rhizosphere soil and roots of *P. tremuloides* and *E. elliptica* from decantation basins with lowest (LC), intermediate (MC) and highest (HC) petroleum hydrocarbon contamination. The identity of each OTU is summarized in Appendix A.

**Figure 3 microorganisms-08-00872-f003:**
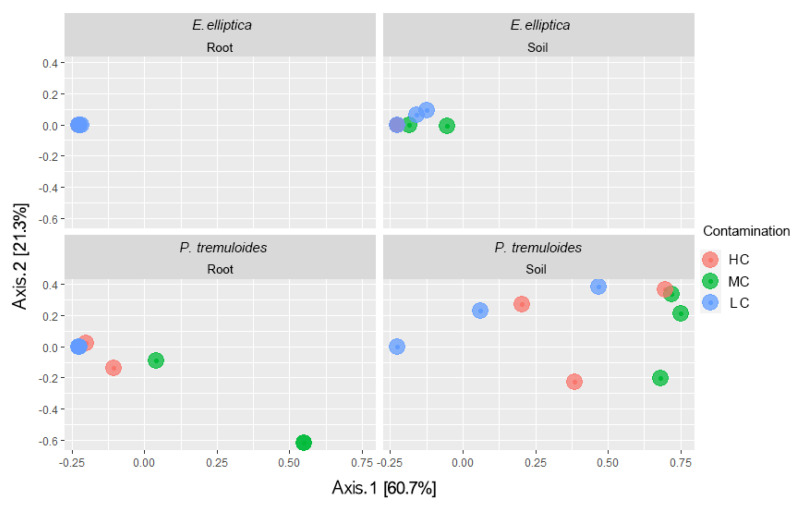
Principal coordinate analysis (PCoA) performed on AMF communities from the rhizosphere soil and roots of *P. tremuloides* and *E. elliptica* from decantation basins with lowest (LC, blue), intermediate (MC, green) and highest (HC, red) petroleum hydrocarbon contamination, based on Bray–Curtis dissimilarity. Axis 1 explains 60.7% of the variation in the community composition, while axis 2 explains 21.3% of the variation. In the samples of *E. elliptica*, overlapping of points existed.

**Figure 4 microorganisms-08-00872-f004:**
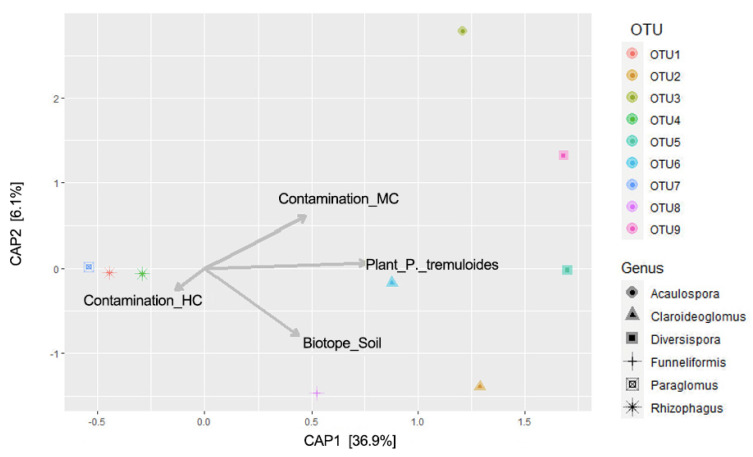
Distance-based redundancy analysis (db-RDA) summarizing the relationship between AMF OTU assemblages with three environmental factors: host plants species, contamination levels and biotopes. The identity of each OTUs were summarized in Appendix A. OTUs assigned to same genus have the same dot shape. A Monte Carlo permutation test with 1000 replicates was applied.

**Table 1 microorganisms-08-00872-t001:** Alpha-diversity of arbuscular mycorrhizal fungi with regards to the host plant species identity and hydrocarbon contamination level in decantation basins from a former petrochemical plant. Mean and standard deviation (S.D.) were calculated from three replicates (*n* = 3).

^1^ Contamination	Plant Species	Biotope	^2^ Shannon		^2^ Simpson		^2^ InvSimpson	
			Mean	S.D.	Mean	S.D.	Mean	S.D.
LC	*E. elliptica*	Root	0.198941	0.199154	0.110179	0.118500	1.137693	0.156639
LC	*E. elliptica*	Soil	0.292518	0.186952	0.151000	0.105701	1.19018	0.148864
LC	*P. tremuloides*	Root	0.237981	0.124852	0.124935	0.080647	1.149346	0.107140
LC	*P. tremuloides*	Soil	0.441605	0.409238	0.279218	0.256173	1.505256	0.506857
MC	*E. elliptica*	Root	0.051359	0.088956	0.022959	0.039766	1.024658	0.042708
MC	*E. elliptica*	Soil	0.431257	0.375213	0.303265	0.264944	1.561379	0.500268
MC	*P. tremuloides*	Root	0.336737	0.583245	0.180979	0.313464	1.39596	0.685822
MC	*P. tremuloides*	Soil	0.731617	0.509229	0.402308	0.271181	2.054042	1.260088
HC	*E. elliptica*	Root	0.055981	0.096963	0.0256	0.044341	1.02773	0.048029
HC	*E. elliptica*	Soil	0.085773	0.148563	0.044218	0.076587	1.05098	0.088301
HC	*P. tremuloides*	Root	0.402262	0.225463	0.209017	0.138528	1.293265	0.248802
HC	*P. tremuloides*	Soil	0.349495	0.605343	0.212032	0.367249	1.582656	1.009189

^1^ Communities from lowest, intermediate and highest contamination basins are presented as LC, MC and HC. ^2^ Shannon diversity index (Shannon); Simpson diversity index (Simpson); inverse Simpson diversity index (InvSimpson).

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
