# Peer review of "Arbuscular Mycorrhizal Fungal Communities of Native Plant Species under High Petroleum Hydrocarbon Contamination Highlights Rhizophagus as a Key Tolerant Genus"

_microorganisms, 2020, doi:10.3390/microorganisms8060872_

Round 1

Reviewer 1 Report

  • Abstract – “extreme levels…” – if we consider rock oil spontaneously leaking out of rock crack – then it can be extreme, nevertheless please avoid using word that can be interpreted in different ways by different people. Just provide exact value for hydrocarbons’ concentration or at least range
  • “The clear association of Rhizophagus with higher contamination levels suggests the importance of the genus for the use of AMF in bioremediation, as well as for the survey of key AMF genes related to petroleum hydrocarbon resistance.” – as a readers I do not see any clear association after reading this particular abstract. Please add details and provide clear interpretation of data. Please be more specific and clearly say what is the relation between bioremediation, resistance etc. Shall we expect that hydrocarbons will be degraded by bacteria and plant together with fungi will support those bacteria ? Or maybe fungi will degrade hydrocarbons supported by plant. The exact message is lost between lines. There should be conclusion or a summary for broader scientific audience at the end of an abstract.
  • Introduction – what is the reason to perform this study ? What is the scientific hypothesis that will be verified in the course of experimental work ? Authors should more clearly highlight the current state of the art and any need to work in this topic.
  • Sampling sites should be briefly described together with plants. I see no justification to choose selected plants. Authors should be more precise and explain in few words the strategy behind their experiments.
  • Soil contamination – if we consider pollution range from few milligrams up to 91 milligrams – I would rather say that the soil is polluted but this is not an extreme environment – please refer to appropriate references and fit your soil within specific classification
  • Results – this section is a bit too long what affects its readability – I would suggest to combine it a little
  • Discussion is generally OK however it would be beneficial to discuss authors’ results in a wider context. What is the general conclusion ?

Author Response

Abstract – “extreme levels…” – if we consider rock oil spontaneously leaking out of rock crack – then it can be extreme, nevertheless please avoid using word that can be interpreted in different ways by different people. Just provide exact value for hydrocarbons’ concentration or at least range

  • Response: we thank the reviewer for this suggestion. We have changed the term “extreme” by “high”. As per Quebec Government’s regulation, the concentration of total petroleum hydrocarbons C10-C50 higher that 3500 mg/Kg is considered as level “C” that characterizes industrial polluted soils. We found a concentration of 91000 mg/Kg in one of the three basins, which is 26-fold higher than C10-C50 polluted soil threshold. We added a sentence in the abstract with the exact value as suggested by the reviewer: (91000 mg/Kg of C10-C50 was recorded in a basin which 26-fold higher that the threshold of polluted soil).

“The clear association of Rhizophagus with higher contamination levels suggests the importance of the genus for the use of AMF in bioremediation, as well as for the survey of key AMF genes related to petroleum hydrocarbon resistance.” – as a readers I do not see any clear association after reading this particular abstract. Please add details and provide clear interpretation of data. Please be more specific and clearly say what is the relation between bioremediation, resistance etc. Shall we expect that hydrocarbons will be degraded by bacteria and plant together with fungi will support those bacteria ? Or maybe fungi will degrade hydrocarbons supported by plant. The exact message is lost between lines. There should be conclusion or a summary for broader scientific audience at the end of an abstract.

  • Response: the Rhizophagus genus was the dominant taxon in AMF communities associated with plants that were sampled in basins where concentration of 91000 mg/Kg petroleum hydrocarbons C10-C150 was recorded. Moreover, our distance-based redundancy analysis revealed that OTUs of this genus (star symbol in Figure 4) were well associated with the highest concentration of petroleum hydrocarbons. Therefore, we think that Rhizophagus occurrence was correlated with high contamination concentration. The term “higher” was inappropriate in the sentence which leads to misunderstanding. We replaced “higher” by “high” and we added additional description of results to support the claim. For the conclusion sentence, we added the description of expected function of the genus in bioremediation with previously reported functions.

Introduction – what is the reason to perform this study ? What is the scientific hypothesis that will be verified in the course of experimental work ? Authors should more clearly highlight the current state of the art and any need to work in this topic.

  • Response: we have added new sentences to explain the rational of our study considering the current state of the art. We further added the hypotheses after the objectives of this research.   

Sampling sites should be briefly described together with plants. I see no justification to choose selected plants. Authors should be more precise and explain in few words the strategy behind their experiments.

  • Response: we think there was a misunderstanding. The justification for choosing selected plants was described in detail in Page 4, L104-106: “…the plant diversity differed between the basins and this led us to focus on plant species co-occuring in the three basins. Therefore, in this study, we choose to sample the following two plants species: Eleocharis elliptica and Populus tremuloides which were present in all three basins, allowing us to compare their related AMF community structures...”

In short, the reasons of choosing these two plant species were: 1) the plants were naturally occurring in the high petroleum contaminated sites in our study. 2) Two plants were only plant species that co-occurred in all three sites, which can allow us to compare the AMF community associated with the host plants growing in different levels of contamination. These reasons are described in the manuscript.  

Soil contamination – if we consider pollution range from few milligrams up to 91 milligrams – I would rather say that the soil is polluted but this is not an extreme environment – please refer to appropriate references and fit your soil within specific classification

  • Response: we have changed the description “extreme” to be “high” through out the manuscript.

Results – this section is a bit too long what affects its readability – I would suggest to combine it a little

  • Response: we understand the reviewer’s concern about the length of the section. We think that it is important to present our results in detail as the current design of study is having three main variables (sites, plant identity and biotopes) which is bit more complicated to be shorten. We afraid that by merging or reducing some paragraphs, readers might get more difficulties to understand. At the same time, the current description of results were well digested by other reviewer. We would like to keep the section length as it is.

Discussion is generally OK however it would be beneficial to discuss authors’ results in a wider context. What is the general conclusion ?

  • Response: we agree. It would have been better to generalize the perspectives of our finding for broader and wider context, but we were afraid of over emphasize what we found. We have put additional effort to expand our conclusion and perspectives. We also added an example of the study outcomes.

Reviewer 2 Report

The manuscript describes a study aimed to verify the AM fungal communities in contaminated soils. The authors used a classical metagenomics approach (cloning and sequencing) using specific primers for the 18S. Results showed that extreme concentration of petroleum-contamination strongly influenced the AM fungal diversity. Although the results are well presented and discussed, I suggest to improve the sampling description as well as to clarify if clones from different replicates were independently sequenced in the M&M section. In the figure of alpha-diversity there is written (n=3), but this point should be clarify in the M&M. At line 122 there is written three replicates, but have the authors considered them as separated samples for sequencing? 

Additionally, at line 122-123 the authors should avoid to cite a plant species that has not been used in this work.

Author Response

The manuscript describes a study aimed to verify the AM fungal communities in contaminated soils. The authors used a classical metagenomics approach (cloning and sequencing) using specific primers for the 18S. Results showed that extreme concentration of petroleum-contamination strongly influenced the AM fungal diversity. Although the results are well presented and discussed, I suggest to improve the sampling description as well as to clarify if clones from different replicates were independently sequenced in the M&M section. In the figure of alpha-diversity there is written (n=3), but this point should be clarify in the M&M. At line 122 there is written three replicates, but have the authors considered them as separated samples for sequencing? 

  • Response: we thank the reviewer for this comment. As described in Mat&Meth, we have sampled 3 individual plants per each species (N=3). For each plant, roots and rhizospheric soil were sampled. Sequencing was based on each biological replicate (each plant, NOT by pseudo sampling of 3 from same sample). We rephrased the description for the clarity.

Additionally, at line 122-123 the authors should avoid to cite a plant species that has not been used in this work.

  • Response: to increase the sample number for composite soil analysis per each site, the soil samples were collected with random sampling of location by which the soils of non-mycorrhizal plant was also included. This does not change the results of composite soil analysis, but it allows to provide description details on the origin of the sample.

Round 2

Reviewer 1 Report

Authors did a good job revising the manuscript.